# A turquoise fluorescence lifetime-based biosensor for quantitative imaging of intracellular calcium

Franka H. van der Linden [1], Eike K. Mahlandt[1], Janine J. G. Arts[1,2], Joep Beumer[3], Jens Puschhof[3], Saskia M. A. de Man [1], Anna O. Chertkova[1], Bas Ponsioen[4], Hans Clevers [3], Jaap D. van Buul [1,2], Marten Postma [1], Theodorus W. J. Gadella Jr.[1] & Joachim Goedhart [1✉]

The most successful genetically encoded calcium indicators (GECIs) employ an intensity or ratiometric readout. Despite a large calcium-dependent change in fluorescence intensity, the quantification of calcium concentrations with GECIs is problematic, which is further complicated by the sensitivity of all GECIs to changes in the pH in the biological range. Here, we report on a sensing strategy in which a conformational change directly modifies the fluorescence quantum yield and fluorescence lifetime of a circular permutated turquoise fluorescent protein. The fluorescence lifetime is an absolute parameter that enables straightforward quantification, eliminating intensity-related artifacts. An engineering strategy that optimizes lifetime contrast led to a biosensor that shows a 3-fold change in the calcium-dependent quantum yield and a fluorescence lifetime change of 1.3 ns. We dub the biosensor Turquoise Calcium Fluorescence LIfeTime Sensor (Tq-Ca-FLITS). The response of the calcium sensor is insensitive to pH between 6.2–9. As a result, Tq-Ca-FLITS enables robust measurements of intracellular calcium concentrations by fluorescence lifetime imaging. We demonstrate quantitative imaging of calcium concentrations with the turquoise GECI in single endothelial cells and human-derived organoids.

[1] Section of Molecular Cytology, van Leeuwenhoek Centre for Advanced Microscopy, Swammerdam Institute for Life Sciences, University of Amsterdam, Amsterdam, The Netherlands. [2] Department of Molecular Hematology at Sanquin Research and Landsteiner Laboratory, Academic Medical Centre, University of Amsterdam, Amsterdam, The Netherlands. [3] Oncode Institute, Hubrecht Institute, Royal Netherlands Academy of Arts and Sciences and University Medical Center, Utrecht, The Netherlands. [4] Center for Molecular Medicine, Oncode Institute, University Medical Centre Utrecht, Utrecht, The Netherlands. ✉email: j.goedhart@uva.nl

Genetically encoded calcium indicators (GECIs) are popular tools for probing intracellular calcium levels[1]. Fierce engineering efforts have led to calcium probes with an impressive intensity contrast, enabling functional imaging in complex tissue and animals[2,3]. The probe design that delivers the largest intensity contrast uses a circular permutated green fluorescent protein (cpGFP) flanked by calmodulin (CaM) and a peptide that binds calcium-bound CaM[3–5]. The calcium-dependent interaction between CaM and the peptide results in a conformation change that is converted into a change in fluorescence intensity of the cpGFP. These probes are also known as GCaMP or GECO[6]. Alternatively, chemical dyes can be used for calcium sensing, but they are limited in application to cellular cultures and they are, unlike GECIs, not targetable to specific organelles[7].

Despite their success, the GCaMP-type probes have limitations. First, the intensity-based readout hinders quantification. The calcium levels modify the fluorescent intensity of the cpGFP. However, fluorescence intensity can also be changed by many additional factors, including photobleaching, sample movement, and changes in (local) probe concentration[8], making it inherently difficult to quantify. Second, GCaMP-type probes are sensitive to pH[9,10] (Supplementary Fig. S1). The binding of calcium changes the p$K_a$ of the cpGFP, which results in a change in the protonation of the chromophore, which ultimately leads to the intensity change[11]. The p$K_a$ of GECIs is near the physiological pH and therefore a change in intracellular pH also changes the intensity. Together, these factors complicate true quantitative imaging of calcium.

Some of these effects can be corrected by ratio imaging using a second fluorescent protein (FP), usually an orange FP (OFP) or red FP (RFP)[12,13]. Unfortunately, the ratio depends on the intensity of the second FP, which in turn is determined by its maturation[12]. In addition, these sensors present the same pH sensitivity as their parents[12,13]. Förster resonance energy transfer (FRET)-based probes are another alternative but have the same issues with unequal maturation rates of the two FPs. Also, the widely used yellow FP (YFP) as acceptor is relatively pH sensitive[14]. Sensitivity for pH and differences in maturation between the FPs results in a variable ratio between cells. In the case of tissue imaging, this is further complicated by wavelength-dependent light scattering[15].

This cell-to-cell variability of the emission ratio hinders quantification and demands establishment of the dynamic range for each individual cell. In addition, intensity and, therefore, ratio imaging are dependent on instrumentation, and data obtained on different microscopes are not comparable. Lastly, FRET and ratio imaging requires a large portion of the visible spectrum, reducing possibilities for multiplexing.

The excited state fluorescence lifetime of a fluorescent molecule is generally not influenced by intensity-related factors[16–18] and can be used successfully for quantitative imaging[19–23]. Consequently, GECI probes based on lifetime contrast would enable true quantitative calcium imaging, independent of equipment. However, most current GECIs, including GCaMPs and FRET-based GECIs, show hardly any or no change in quantum yield (QY) or lifetime contrast (Supplementary Fig. S1), despite a clear intensity change[24,25]. For the FRET sensors, this is caused by a close to 100% efficiency in energy transfer in one of the states of the sensor. In this state, the donor fluorescence is completely quenched; hence, no photons are emitted and no donor lifetime can be measured. However, FRET can still be observed by ratio imaging, as the energy is fully transferred to the acceptor, increasing its fluorescence. In cpGFP-based sensors, the intensity change is predominantly caused by a change in extinction coefficient ($\varepsilon$), without a significant change in QY[3,24] (Supplementary

Fig. S1). This leads to an absence of lifetime change, as the lifetime of FPs is proportional to the QY.

Exceptions are R-GECO1[6], K-GECO1[26], RCaMP1h, and jRCaMP1b[27], which are intensity-based calcium sensors that employ a RFP instead of a GFP, and they show a QY contrast. For RCaMP1h and jRCaMP1b, a corresponding lifetime contrast has been shown[27,28]. However, these probes all display pH sensitivity in the biological range. In addition, they have a very low intensity in the calcium-free state, complicating lifetime measurements (Supplementary Fig. S1).

In this study, we first verify the performance of several of the calcium sensors discussed above. Then we show the development of a GECI, which does display a robust calcium-dependent lifetime contrast, with sufficient brightness to accurately measure calcium concentrations across its full dynamic range, and with a low sensitivity for pH. Finally, we demonstrate the advantageous features of this sensor by measuring the calcium concentration in endothelial cells (ECs), in a model for migration of white blood cells through the blood vessel wall, and by detecting the calcium concentration in human intestinal organoids.

## Results

**Overview of existing calcium sensors**. We started out by verifying the performance of existing calcium sensors in order to establish their potential for quantitative calcium measurements. Intensity of a single FP is inherently unsuitable for quantitative measurements, as intensity is influenced by many factors including movement, concentration, bleaching, and sample thickness. Therefore, this mode of readout was excluded. First, we looked into ratiometric sensors, either based on FRET imaging or on the ratio between a cpFP calcium sensor and a second FP. We chose YCaM3.60 as example of a FRET sensor and Matryosh-CaMP6s (consisting of GCaMP6s fused to LSSmOrange) as example of a ratio sensor. We observed for both types of ratiometric sensors a high variability in emission ratio between cells (Fig. 1 and Supplementary Table S1). This large variability was observed in both unstimulated HeLa cells and in cells stimulated with ionomycin, a drug that increases the intracellular calcium concentration. We plotted the ratio normalized to the initial frames during stimulation with ionomycin (Fig. 1a). Again, for both sensors, a large variability was observed. All experiments were carried out on two different microscopes, showing the influence of the instrumentation on the ratios measured. For MatryoshCaMP6s, we noted a negative correlation between the intensity of the LSSmOrange and the measured GFP/OFP ratio (Fig. 1b).

Next, we looked into the possibility of FRET-fluorescence lifetime imaging microscopy (FLIM) measurements, as lifetime is a more robust readout for the conformational state of a FRET probe than intensity[29]. YCaMP3.60 shows a clear response in donor intensity upon ionomycin addition, indicative of FRET; however, we observed only a very minor donor lifetime change (Supplementary Fig. S2). We tested three other calcium FRET sensors of the Twitch series, leading to the same results. In contrast, a dedicated FRET-FLIM biosensor[30] for cyclic AMP (cAMP) indeed shows an intensity and concomitant lifetime change upon stimulation with forskolin, a drug that stimulates cAMP production.

Finally, we looked at RCaMP1h and jRCaMP1b for which previously a lifetime change between the on- and off-state has been reported. We were able to measure a substantial lifetime change in HeLa cells upon stimulation with ionomycin, with little variation (Supplementary Table S1).

Of all investigated existing sensors only the RCaMPs show a decent lifetime contrast that can be used as a quantitative readout.

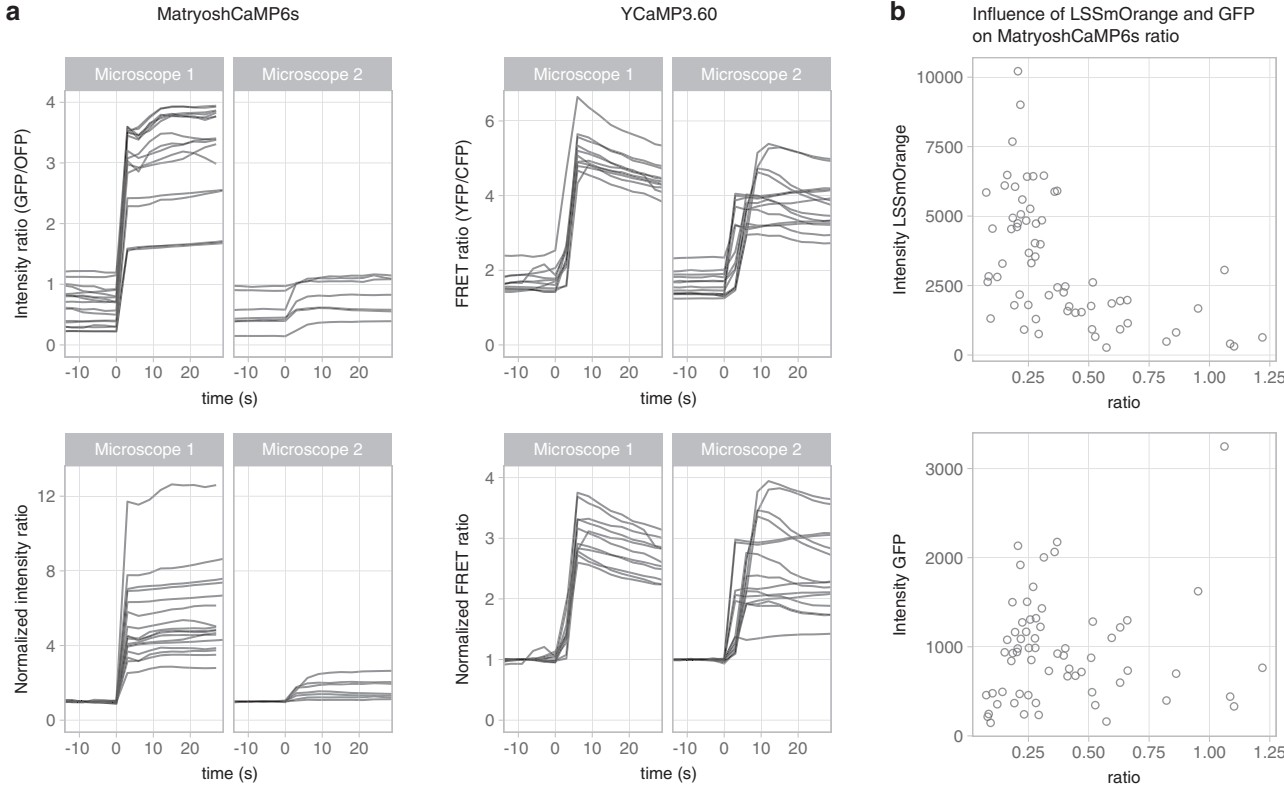

**Fig. 1 Cell-to-cell variability in ratiometric measurements of MatryoshCaMP6s and YCaM3.6O in HeLa cells. a** Time series of individual cells stimulated at $t = 0$ with 14 mM ionomycin combined with 5 mM $CaCl_2$ ($n = 7$ to 17), taken on two different microscopes: a NIKON (1) and a ZEISS (2) setup. Top panels show the intensity ratio, bottom panels show the ratio normalized to the first three frames. Both ratio and normalized ratio give a highly variable output. **b** The intensity of LSSmOrange of MatryoshCaMP6s (top panel) shows a clear influence on the measured ratio, which is not the case for the green fluorescence (bottom panel). Dots indicate individual cells ($n = 64$), data for **b** were acquired with microscope 1. Source data are provided as a Source Data file.

We reasoned that an ideal quantitative calcium sensor would also have limited to no sensitivity to pH in the biological range. In case of lifetime imaging, a good intensity in both the calcium-bound and -unbound state is desirable. The currently available sensors do not meet all these criteria, so we set out to develop a sensor that displays lifetime contrast.

**Engineering a turquoise lifetime sensor**. We selected cyan fluorescent protein (CFP) mTurquoise2 as the FP to generate a calcium sensor, as it has a high QY, high lifetime, and we have a thorough understanding of residues that affect its QY and lifetime[31]. In addition, the $pK_a$ of mTurquoise2 is low[32] (~3.6) and it is unlikely that a conformation change would affect the protonation state and $\varepsilon$.

Following the approach that led to the GECO series[6], CaM and its binding peptide M13 were attached to the N and C termini of a circular permutated mTurquoise2 (cpTq2). As information on circular permutated cyan FPs is hardly available, we experimentally determined the optimal site for circular permutation of mTurquoise2. To this end, cpTq2 sensor variants were cloned into a dedicated dual expression vector termed pFHL, which allows protein expression in mammalian cells and *Escherichia coli*, and is suitable for isolation of the protein of interest. In addition, the protein of interest is partially transported to the periplasm of bacteria, allowing for a quick and crude extraction (Supplementary Note 1 and Supplementary Fig. S3). Nine candidate sensors were constructed: seven with the termini in the seventh β-strand and the remaining two in the tenth β-strand of the mTurquoise2 β-barrel (Fig. 2a). Seven of the nine candidate

sensors showed fluorescence in *E. coli* and HeLa cells. The sensor with the CaM and M13 attached to amino acids 149 and 150, respectively (which we named Turquoise Calcium Fluorescence LIfeTime Sensor version 0 (Tq-Ca-FLITS.0)), showed about a twofold change in intensity in periplasmic fluid isolated from bacteria upon addition of calcium (periplasm test, Supplementary Fig. S4a), whereas the others showed little to no response. When expressed in HeLa cells, Tq-Ca-FLITS.0 showed a threefold intensity change upon addition of ionomycin (Fig. 2b and Supplementary Fig. S4b). We recorded the lifetime in both states of all seven candidate sensors using frequency domain FLIM. Tq-Ca-FLITS.0 again showed the largest response, with a striking phase and modulation lifetime change of over 1 ns (Fig. 2b, c and Supplementary Fig. S4c). The other variants showed only marginal changes in lifetime.

Next, we set out to improve the lifetime contrast of the sensor by creating variants with up to two insertions or five deletions on both sides of the cpTq2. Many variants with the cpTq2 starting with amino acid V150 showed an intensity fold-change in the periplasmic test and in HeLa cells. However, the original variant without indels still showed the greatest lifetime contrast (Supplementary Fig. S4). Tq-Ca-FLITS.0 was therefore subjected to mutagenesis on two key residues that affect the fluorescence lifetime, i.e., F146 and V150 (original mTq2 numbering). F146 was previously shown to have a large influence on the lifetime of mTurquoise2[31]. V150 was selected for its position with respect to the chromophore, and in a small screen it showed to have an influence on the lifetime of mTurquoise2 (Supplementary Note 2 and Supplementary Fig. S5). A library of Tq-Ca-FLITS.0

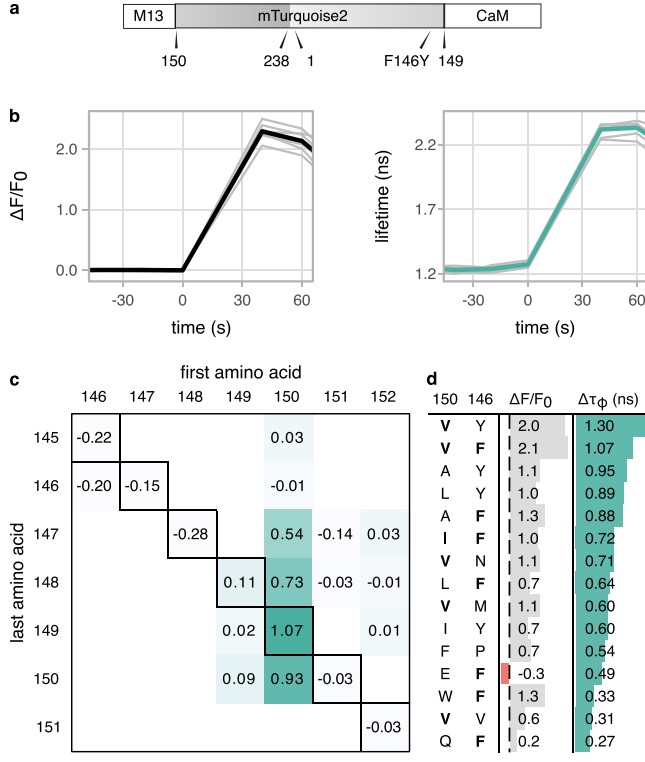

**Fig. 2 Development of the Turquoise calcium sensor. a** Schematic overview of the layout of Tq-Ca-FLITS, including key mutation F146Y. Positions of the original (amino acid 1 and 238) and the new (150 and 149) N and C termini are indicated. **b** Intensity fold-change over $F_0$ and phase lifetime response of Tq-Ca-FLITS.0 in HeLa cells stimulated with ionomycin and calcium at $t = 0$. Responses of individual cells (gray, $n = 7$) and their mean (black or green) are plotted. **c** Absolute phase lifetime change of various sensor variants in HeLa cells stimulated with ionomycin and calcium. The first amino acid of mTurquoise2 after the M13 peptide and the last amino acid before the CaM are indicated. On the diagonal axis (black outline) are the sensor variants that were created to find the ideal position to insert the CaM and the M13 peptide. The other variants contain two to four indels around the insertion site. Values are means of 2–23 individual cells. **d** Intensity fold-change over $F_0$ and absolute phase lifetime change of mutants of Tq-Ca-FLITS.0 in HeLa cells stimulated with ionomycin and calcium. Amino acids at position 150 and 146 of mTurquoise are indicated. The original residues (V and F) are printed in bold. Values are means of two to ten individual cells. Source data are provided as a Source Data file.

containing F146X and/or V150X mutation(s) was initially screened in *E. coli* on agar plates, selecting for colonies showing a high intensity. We noticed that the sensors under these conditions are primarily in the high lifetime and high intensity state (Supplementary Note 3 and Supplementary Fig. S6). By selecting the high lifetime colonies, we aimed to increase the overall brightness of the probe and increase the lifetime of the calcium-bound state. Over 450 colonies were screened of which 60 "high" intensity colonies were selected and 17 "low" and 16 "intermediate" colonies as control. The selected colonies were screened in liquid bacterial culture, by monitoring the change in fluorescence intensity upon addition of calcium chelator EDTA to the culture. The periplasmic fluid of the best-performing candidates was isolated for further testing (with the periplasm test) and their DNA sequence was determined. Optimal responses were observed when amino acids at position 150 were V or A, and F or Y at position 146 (Supplementary Fig. S7). Based on these results, five additional variants were constructed, with I, L, or A at position 146 and F or Y at position 150. The designed variants

and the top candidates from the screen were tested in HeLa cells for lifetime contrast (Fig. 2d). We identified a variant with a F146Y mutation that showed a comparable intensity response as the original variant and an increased phase lifetime response of about 1.3 ns. This variant was named Tq-Ca-FLITS.

**Characterization of Tq-Ca-FLITS**. In vitro characterization of Tq-Ca-FLITS showed a substantial difference in QY between the calcium-bound and calcium-free state (75% and 25%, respectively), which is in line with the lifetime change. The extinction coefficient between the two states is comparable, unlike virtually all "GCaMPs"[3,24] (Table 1, Supplementary Fig. S2, Fig. 3a, and Supplementary Fig. S8).

We investigated whether the lifetime readout of Tq-Ca-FLITS was influenced by the pH. Strikingly, the lifetime readout of Tq-Ca-FLITS is insensitive to pH above 6.2, making it a robust probe for quantitative biological measurements (Fig. 3b and Supplementary Fig. S9) and ideally suited for in situ calibration[33]. To this end, we established a HeLa cell line with stable expression of the sensor in the nucleus. Cells were incubated in calcium buffers and the calcium concentration in the cytoplasm was equilibrated with the outer environment by permeabilization of the membrane with digitonin. The concentration of calcium is plotted against the lifetime that was measured when equilibrium was reached (Fig. 3c, d and Supplementary Movie S1). We converted the phase and modulation lifetime values of all calibration experiments to polar coordinates[34], which resulted in a straight line on a polar (or phasor) plot (Supplementary Fig. S10a). For each calcium concentration, we calculated the fraction of the sensor in the high lifetime state. This resulted in an apparent $K_d$ of 265 nM, which is comparable to the GCaMP6 series[35] (Table 1, Supplementary Table S3, and Supplementary Fig. S10b). The apparent $K_d$ in vitro was determined to be 372 nM, using isolated Tq-Ca-FLITS and applying a similar calculation on the lifetime data as for the in situ calibration (Supplementary Table S3 and Supplementary Fig. S10c, d). When using the intensity data of the in vitro calibration, we found a similar apparent $K_d$ of 360 nM. This is comparable to the apparent $K_d$ in vitro of R-GECO1, the parent of Tq-Ca-FLITS[6], and to jGCaMP7c (Table 1, Supplementary Table S2, and Supplementary Fig. S10e, f). Using the variation of the in situ calibration, we determined the detection range of the sensor to be 20 nM–1.8 µM (Supplementary Fig. S10b).

Finally, we compared the pH and magnesium sensitivity of Tq-Ca-FLITS to jGCaMP7c and RCaMP1h. In line with the lifetime, also the intensity readout of Tq-Ca-FLITS is unhampered by pH above 6.2, showing a stable dynamic range of 3.5 (Fig. 4). Using the intensity data, we determined a $pK_{a,apo}$ of 4.4 (Hill coefficient 0.86) for the free, unbound state of Tq-Ca-FLITS. A model with two $pK_a$ values was used for the calcium-bound state and resulted in a $pK_{a,sat,1}$ of 4.7 (Hill coefficient 0.70) and a $pK_{a,sat,2}$ of 5.9 (Hill coefficient 3.6) (Table 1 and Supplementary Table S2). The low and very similar $pK_{a,apo}$ and $pK_{a,sat,1}$ are likely a direct result of the pH dependency of the FP in Tq-Ca-FLITS. The $pK_{a,sat,2}$ probably shows the pH sensitivity of the four calcium binding domains of the CaM, which is also supported by the high Hill coefficient[36]. This $pK_{a,sat,2}$ corresponds with the isoelectric point of CaM binding domains, which is shown to be around 6[37]. As expected, jGCaMP7c and RCaMP1h show a clear pH sensitivity in the biological range (jGCaMP7c: $pK_{a,sat} = 6.8$ and $pK_{a,apo} = 9.1$; RCaMP1h: $pK_{a,sat} = 6.2$ and $pK_{a,apo} = 7.3$).

We found a very low magnesium sensitivity for Tq-Ca-FLITS, both in intensity and lifetime readout (Supplementary Fig. S11). However, jGCaMP7c does show a marked change in dynamic range at the low magnesium concentrations (below 1 mM), which is reported to be right in the biological range[38].

**Table 1 Properties of Tq-Ca-FLITS.**

| | Spectral properties | | | | | Intensity data | | | Lifetime data | | |
|---|---|---|---|---|---|---|---|---|---|---|---|
| | $\lambda_{abs}$ (nm) | $\lambda_{em}$ (nm) | QY | $\varepsilon$ (mM$^{-1}$ cm$^{-1}$) | p$K_a$ [n] | $K_d$ [n] (nM) pH 7.2 | $F_{max}/F_{min}$ in vitro pH 7.0 | $F_{max}/F_0$ in situ | $\tau_\varphi$ (ns) | $\tau_M$ (ns) | $K_d$ [n] (nM) |
| Apo | 442 | 489 | 0.25 | 30.6 | 4.36 [0.86] | 360 [1.51] | 3.51 | 3.02 | 1.40 | 1.80 | 265 [1.63] |
| Sat | 439 | 481 | 0.75 | 33.7 | 1: 4.71 [0.70] 2: 5.91 [3.58] | | | | 2.78 | 3.01 | |

Spectral properties: $\lambda_{abs}$ absorbance maximum; $\lambda_{em}$ emission maximum; QY quantum yield relative to mTurquoise2; $\varepsilon$ extinction coefficient at 440 nm; p$K_a$ [n] apparent p$K_a$ value, with the Hill coefficient [n] between brackets. A model with two p$K_a$ values was used for the calcium-saturated state.
Intensity: $K_d$ [n] apparent $K_d$ in vitro, with the Hill coefficient [n] between brackets; $F_{max}/F_{min}$ ratio of maximum over minimum fluorescence intensity, or fluorescence of calcium-bound state over calcium-unbound state; $F_{max}/F_0$ maximum intensity over starting intensity in situ, as determined by stimulation with ionomycin and calcium.
Lifetime: quantitative in situ readout with $\tau_\varphi$ phase lifetime; $\tau_M$ modulation lifetime; $K_d$ [n] apparent $K_d$, with the Hill coefficient [n] between brackets, determined from the position of the lifetimes on a polar plot.

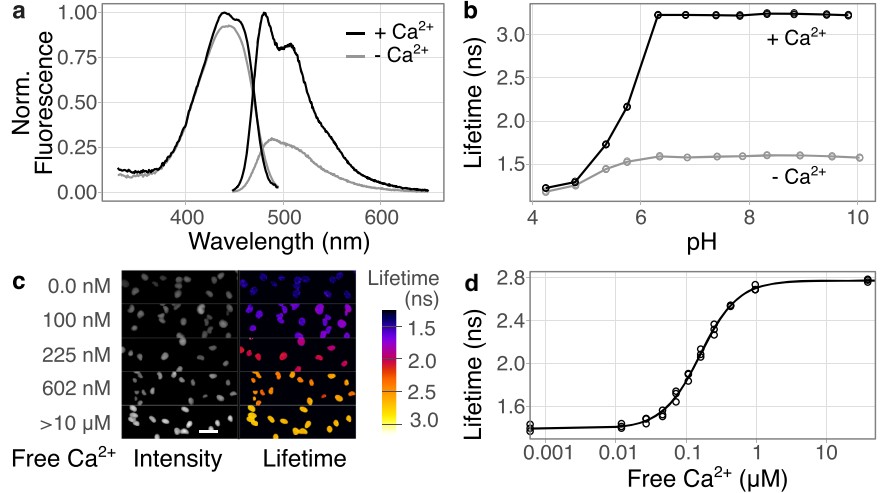

**Fig. 3 Main characteristics of Tq-Ca-FLITS. a** Normalized absorbance and emission spectra of Tq-Ca-FLITS in vitro for the calcium-bound and -unbound state. **b** Tq-Ca-FLITS is pH insensitive from pH 6.2 onwards, as shown by the phase lifetime of the calcium-bound and -unbound states in vitro ($n = 3$ with line average). **c** The phase lifetime of Tq-Ca-FLITS stabilizes in HeLa cells during the in situ calibration. The endpoint of five concentrations is shown. Visual display is generated by an ImageJ macro[52]. Scale bar is 50 μm. **d** Calibration curve of the phase lifetime in situ, with the calcium concentration on a logarithmic scale ($n = 3$). Source data are provided as a Source Data file.

**Measuring the calcium concentration in situ using Tq-Ca-FLITS.** To enable detection in various cellular compartments, we generated probes that localize in the cytoplasm, at the Golgi apparatus, at the plasma membrane (Supplementary Fig. S12) or in the nucleus (Supplementary Fig. S13). Using the nuclear localized variant, we measured the lifetime response in the nucleus of HeLa cells upon stimulation with 2 μM histamine, with a time resolution of 1.6 s (Supplementary Fig. S13 and Supplementary Movie S2). With this acquisition speed, we were able to follow a fast increase of calcium from a baseline of 70–100 nM to more than 300 nM within 20 s.

Next, we examined the performance of the Tq-Ca-FLITS probe for quantitative intracellular calcium imaging in a number of biological systems. It has been well-established that primary ECs respond to histamine with a transient intracellular calcium release[39]. We quantified the calcium levels after stimulation with 1 μM histamine and observed spatial heterogeneity of the calcium distribution, proving that Tq-Ca-FLITS is suitable for local quantification of intracellular calcium (Fig. 5a, b and Supplementary Movie S3).

We have previously shown that ECs actively prevent local leakage from blood vessels when leukocytes cross the endothelial barrier during transendothelial migration (TEM), by inducing a RhoA-dependent F-actin ring that serves as an elastic strap[40].

Whether calcium is involved in the subsequent pore closure is unknown. Although a role for calcium in TEM has been proposed, only the adhesion phase has been studied in some detail, with varying results[41]. The majority of studies used organic dyes with ultraviolet excitation and under non-physiological conditions, i.e., under the absence of flow (Supplementary Note 4 and Supplementary Table S4). We regarded conventional intensity-based GECIs unsuited for quantifying local calcium concentrations due to morphological changes of the cells, leading to substantial intensity changes unrelated to calcium concentrations.

We used a model system for TEM that utilizes flow to mimic physiological conditions, similar to the system we used previously to study RhoA activity[40], but with improvements: (i) we used FLIM in combination with Tq-Ca-FLITS to measure a quantitative output and (ii) we simultaneously imaged fluorescence of the labeled leukocytes, enabling us to precisely correlate the different phases of TEM to the calcium concentration.

We achieved a temporal resolution of 13.5 s, which is sufficient to analyze calcium levels during TEM. Changes in the fluorescence intensity of both the ECs and leukocytes (neutrophils) were observed (Fig. 5c, d and Supplementary Movie S4). The leukocyte intensity is affected by cell shape, as we are using a widefield microscope. High fluorescence intensity corresponds to

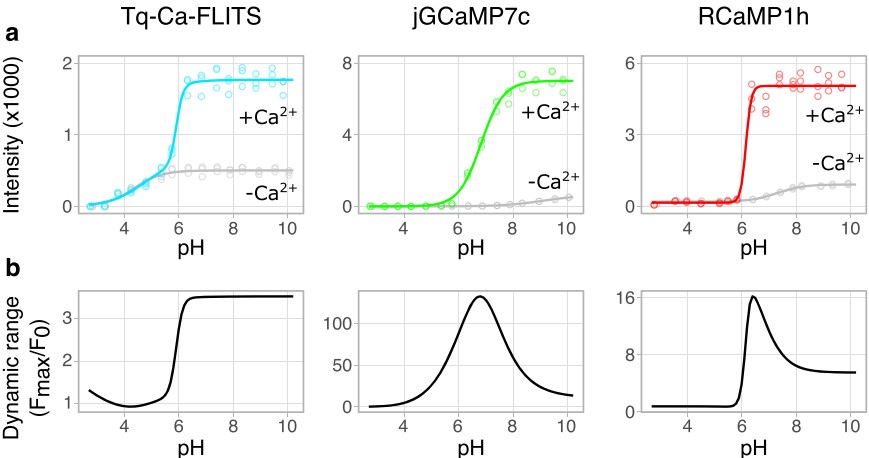

**Fig. 4 pH sensitivity of Tq-Ca-FLITS, jGCaMP7c and RCaMP1h in vitro. a** Intensity change of the sensors as a response to pH, for the calcium-bound (blue/green/red) and -unbound (gray) state (all $n = 3$). A Hill curve (line) with one $pK_a$ value and Hill coefficient was fitted through the measured data (circles), except for calcium-bound state of Tq-Ca-FLITS where a model with two $pK_a$ values was used. **b** The dynamic range is calculated as a ratio of the models presented in **a**. The dynamic range of Tq-Ca-FLITS is stable in the biological pH range, whereas it fluctuates strongly for jGCaMP7c and RCaMP1h. Source data are provided as a Source Data file.

a ball-like shape when the neutrophils are crawling and low intensity is observed when the neutrophils spread out after completing diapedesis. The intensity of Tq-Ca-FLITS in the ECs was also affected by cell shape, and changes were observed when leukocytes continued to migrate under the EC monolayer. In contrast to the intensity changes, hardly any or no lifetime changes of the calcium probe were observed before adhesion, during crawling, or during and after diapedesis of the leukocyte. This translates to hardly any or no calcium changes during TEM (Fig. 5e and Supplementary Movie S4). We analyzed a total of 16 tracks, capturing 98 crawling events and 19 diapedesis events. Also, the calcium concentration before adhesion ($n = 216$) and post-diapedesis ($n = 278$) was determined. The calcium levels in individual cells in almost all these events did not exceed 80 nM. This concentration is comparable to calcium concentrations reported in resting EC monolayers earlier[41–43]. In contrast, in the same events we did observe an intensity fold-change post-diapedesis (Fig. 5e). The addition of ionomycin or histamine to the samples after the TEM assay showed a strong increase in intensity, lifetime, and calcium concentration (>1.5 µM), demonstrating that the calcium probe was fully functional in the experimental context (Supplementary Fig. S14). Our experimental approach that quantifies the calcium concentration in ECs at different stages of TEM shows that calcium elevation (>0.08 µM) in ECs is not essential for pore closure. Next to this, our data suggests that efficient crawling of neutrophils does not require an increase in baseline calcium levels.

**Measuring calcium concentrations in human organoids using Tq-Ca-FLITS.** Finally, the probe was used to quantify the calcium concentration in a human small intestinal organoid[44]. A nuclear-targeted Tq-Ca-FLITS variant was used to simplify the identification of individual cells. We previously observed an intensity change of the Tq-Ca-FLITS probe in response to stimulation of organoids with an odorant; however, lifetime imaging was not performed in that study[44]. Here we investigated whether lifetime changes can be observed in this complex multicellular system. We differentiated organoids to the hormone-producing enteroendocrine cells (EECs), which express multiple G-protein coupled receptors (GPCRs) that control secretion of their products[44]. In vivo, these cells represent <1% of the epithelium, but these can be enriched up to >50% in organoids. Calcium elevation is generally

coupled to release of hormones from these cells. Lifetime imaging of the Tq-Ca-FLITS probe in nuclei of organoid cells revealed an intensity increase and a concomitant calcium increase in several cells when GPBAR-A was added. This drug is an agonist for GPBAR1, a GPCR that is expressed mainly by GLP-1-producing L-cells, a subtype of EECs, present in the organoid (Fig. 5f, g and Supplementary Movie S5). We observed a calcium increase by addition of GPBAR-A from 40–80 nM to an elevated baseline of 70–130 nM with occasional spikes up till 250 nM. As a control, we saturated the calcium sensor in the organoid by addition of Triton X-100 and calcium to the medium (Supplementary Fig. S15).

## Discussion

FPs have been used to develop genetically encoded calcium sensors for the past 20 years. Most efforts have been focused on qualitative detection of calcium changes, but barely on measuring the absolute concentration. To enable the quantitative detection of calcium concentration in cells, we have rationally engineered a genetically encoded biosensor that displays a fluorescent lifetime that depends on the calcium concentration. As the fluorescent lifetime is an absolute parameter, the Tq-Ca-FLITS biosensor enables absolute calcium measurements by lifetime imaging and is expected to simplify calcium imaging, screening, and comparison of results between different microscopes and laboratories.

Our results demonstrate that a circular permutated variant of mTurquoise is a viable template for engineering sensors that (i) show lifetime contrast and (ii) are not sensitive to changes in the cytoplasmic pH range. The use of fluorescence lifetime to quantify calcium concentrations makes the measurements largely insensitive to changes in intensity. This simplifies the calibration in cells and results in a robust in situ calibration. The lower calcium sensitivity of Tq-Ca-FLITS in situ compared to in vitro is most likely caused by environmental differences. This demonstrates the need of an in situ calibration when attempting quantification of intracellular concentrations. Unlike other GECIs, the Tq-Ca-FLITS is not optimized for intensity contrast and is therefore not the probe of choice for binary detection of calcium events. Instead, we have optimized this GECI for lifetime contrast, allowing simple, robust, and precise quantification of intracellular calcium concentrations. The low pH sensitivity of

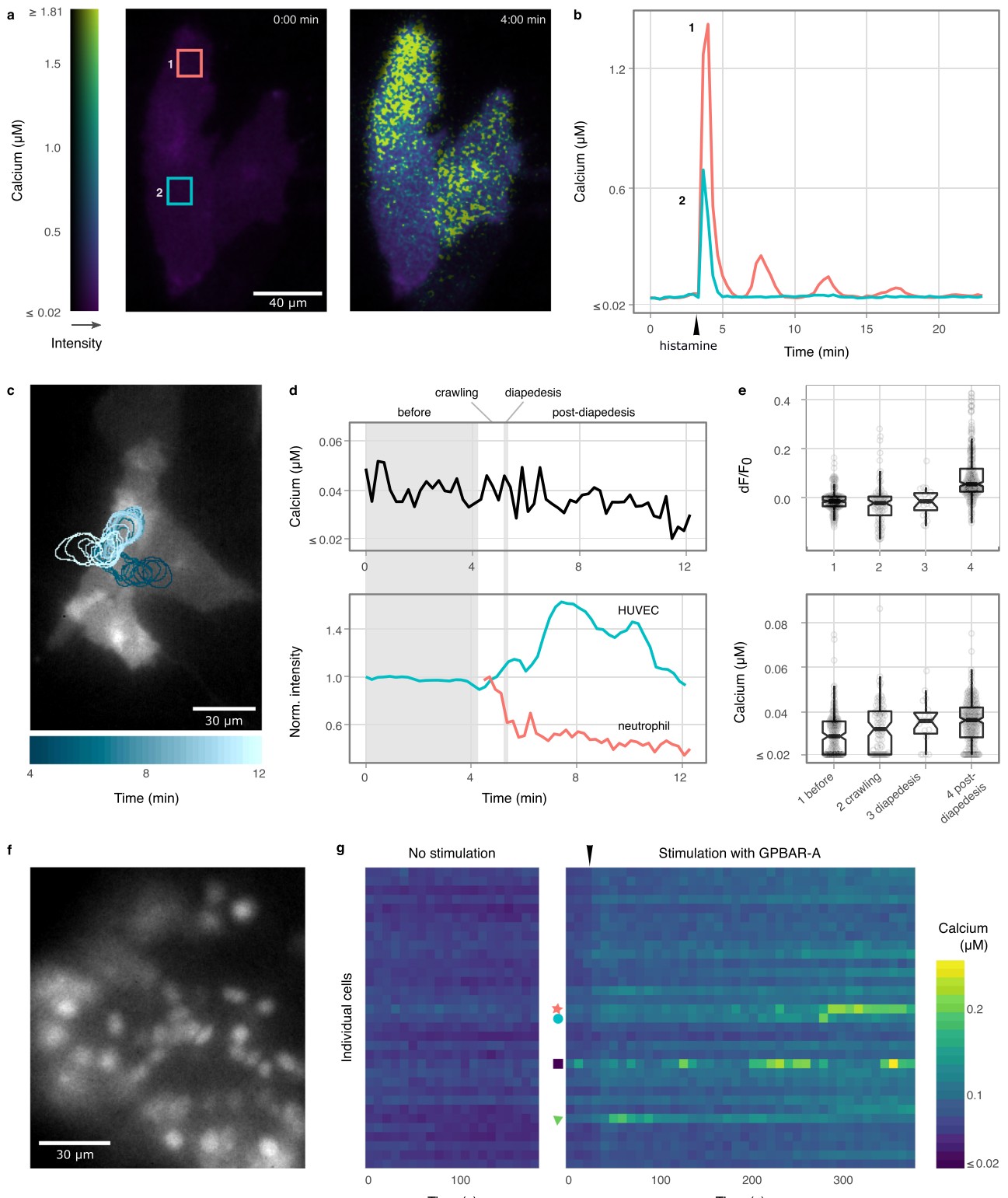

Tq-Ca-FLITS combined with a quantitative readout by means of lifetime is not met by other available sensors.

We demonstrate accurate calcium imaging even when intensity fluctuations are caused by morphological changes, as observed during TEM. The temporal resolution was sufficient to capture the transmigration of neutrophils; however, it could be greatly improved by choice of microscope. Our setup is not suitable for fast switching of the filters required for alternating imaging of cyan lifetime and red fluorescence, and it suffers from substantial

response and dead times. In the choice of setup for our experiments, we sacrificed speed for accurate imaging of several ECs in one field of view. When imaging only the lifetime in the CFP channel, we reached a temporal resolution with Tq-Ca-FLITS of 1.6 s. Lifetime imaging can be further pushed to sub-second resolution by choice of improved and faster lifetime microscopy techniques such as a FALCON or Stellaris systems[45], or siFLIM[46]. The relative intrinsic brightness of Tq-Ca-FLITS is 25–75% ("off"- and "on"-state, respectively) of mTurquoise2, the brightest

**Fig. 5 Quantitative calcium measurements with Tq-Ca-FLITS in primary endothelial cells and human organoids. a** Calcium levels in ECs with plasma membrane targeted Tq-Ca-FLITS before and after stimulation with 1 µM histamine, indicated by an arrow. The color scale indicates the concentration of calcium and the fluorescence intensity. Scale bar is 40 µm, a representative of 17 cells is shown. **b** Time trace of the regions indicated in **a**. **c–e** Calcium concentration in ECs during TEM. The ECs express plasma membrane targeted Tq-Ca-FLITS and the leukocytes are labeled with Calcein Red-Orange. In total, 16 TEM events were captured within seven measurements, using four batches of neutrophils. **c** Representative fluorescence image of the ECs with the biosensor. The blue outline indicates the location of a neutrophil that crosses the EC monolayer. Scale bar is 30 µm. **d** Top panel, calcium concentration in ECs measured at the location of the neutrophil indicated in **c** and bottom panel, the fluorescence intensity of Tq-Ca-FLITS and the leukocyte at the same location. The intensity of the neutrophil drops right after crossing the EC monolayer, whereas the intensity of Tq-Ca-FLITS increases. The concentration of calcium in the EC remains constant, as determined from the fluorescence lifetime of Tq-Ca-FLITS. **e** Intensity fold-change ($dF/F_0$) of Tq-Ca-FLITS (top panel) and measured calcium concentration (bottom panel) in ECs before neutrophil contact ($n = 216$), during crawling of the neutrophil ($n = 98$), during diapedesis ($n = 19$) and post-diapedesis ($n = 278$). The boxplot indicates the median, the 95% CI (notches), the first and third quartiles (hinges), and the 1.5× interquartile range (whiskers). All data points are also indicated by circles. **f–g** Calcium changes measured in cells of a human small intestinal organoid stimulated with 10 µg/ml GPBAR-A. **f** A human small intestinal organoid expressing Tq-Ca-FLITS in the nuclei of the cells. Scale bar is 30 µm. Three organoids showed transient calcium increases, shown is a representative organoid. **g** Heatmap of time traces of the calcium concentration of the cells in **f**, without (left panel) and with (right panel) stimulation by GPBAR-A, indicated by an arrow. Markers indicate the corresponding cells in **f**. Source data are provided as a Source Data file.

cyan FP available. Moreover, in the "off"-state the intrinsic brightness of the Tq-Ca-FLITS is ~76% of ECFP[47], which is still commonly used in FRET ratio probes for timelapse imaging. Therefore, we expect that Tq-Ca-FLITS is suitable for faster and dynamic imaging of calcium concentrations than is demonstrated here. Another promising method for speeding up lifetime measurements is by using a hybrid method with intermittent calibration proposed by Dedecker and colleagues[48].

The experiments with organoids show that we can accurately measure calcium levels in a complex three-dimensional tissue. Human organoids are widely used for modeling of epithelial physiology. Previous work has highlighted—based on single-cell transcriptomic analysis—the near-identical nature of organoid cells compared to their tissue counterparts[49]. The cellular heterogeneity of organoids may well facilitate quantifying cell type-specific calcium responses in a physiologically relevant context. We chose to assess calcium dynamics in human EECs, which eventually control hormone secretion. Human EECs differ greatly from their murine counterparts in terms of their expression profile of GPCRs, and therefore organoids represent a unique model to study EEC functioning in man. These endocrine cells represent important potential targets for treatments of metabolic diseases, as their hormones are involved in controlling key physiological processes such as appetite and insulin secretion. To increase the accuracy of measurements in thick samples, optical sectioning is necessary. This can be achieved by combing lifetime imaging with confocal scanning, light sheet imaging, or multiphoton excitation.

Imaging with blue excitation light generally increases the risk of phototoxicity, especially a problem for in vivo research, but we note that mTurquoise2 has been successfully used for imaging of developing *Nematostella vectensis* embryos on an ordinary confocal microscope[50]. In addition, mTurquoise2 was imaged to study the cell cycle, a process sensitive to photodamage[51]. Together, these observations suggest that probes based on mTq2 can be successfully used in vivo. The use of light sheet imaging or multiphoton excitation could, in addition to improving the aforementioned optical sectioning, reduce the risk of phototoxicity. Furthermore, the blue-shifted color of Tq-Ca-FLITS could be beneficial for multicolor imaging. For example, we have successfully used it in combination with mClover and mTomato[44].

To conclude, Tq-Ca-FLITS is a unique GECI that incorporates a sensing mechanism based on a conformational change that directly modifies only the fluorescence QY and fluorescence lifetime of a FP independent of FRET (and without the need for a second FP), providing contrast independent of sensor concentration. Tq-Ca-FLITS has a calcium sensitivity comparable to other existing GECIs and shows no sensitivity to pH in the biological range. Together, these properties make the sensor ideally suited for quantitative determination of calcium concentrations. We anticipate that this sensor design can be easily combined with other sensor domains, e.g. reporting on phosphorylation and small molecule- or protein-binding, to generate a unique class of fluorescence biosensors for the quantitative analysis of cellular processes.

## Methods

**General cloning**. We used *E. coli* strain *E. cloni* 5-alpha (short: *E. cloni*, Lucigen corporation) for all cloning procedures. For DNA assembly, competent *E. cloni* was transformed using a heat shock protocol according to manufacturers' instructions. For protein expression, *E. cloni* was grown using super optimal broth (0.5% (w/v) yeast extract, 2% (w/v) tryptone, 10 mM NaCl, 20 mM MgSO$_4$, 2.5 mM KCl) supplemented with 100 µg/ml kanamycin and 0.2% (w/v) rhamnose. For agar plates, 1.5% (w/v) agar was added. Bacteria were grown overnight at 37 °C. Plasmid DNA was extracted from bacteria using the GeneJET Plasmid Miniprep Kit (Thermo Fisher Scientific) and the obtained concentration was determined by Nanodrop (Life Technologies).

DNA fragments were generated by PCR, using Pfu DNA polymerase (Agilent Technologies) unless otherwise indicated. DNA fragments were visualized by gel electrophoresis on a 1% agarose gel, run for 30 min at 80 V. PCR fragments were purified using the GeneJET PCR purification Kit (Thermo Fisher Scientific) and digested with restriction enzymes to generate sticky ends. Restriction enzymes were heat inactivated at 80 °C for 20 min if necessary. Vector fragments were generated by restriction of plasmids and the correct bands were extracted from gel using the GeneJET Gel Extraction Kit (Thermo Fisher Scientific).

DNA fragments were ligated using T4 DNA ligase (Thermo Fisher Scientific), per the manufacturers' protocol. Correct construction of plasmids was verified by control digestion and sequencing (primers 38–39, Macrogen Europe). All primers (Supplementary Table S5) were ordered from Integrated DNA Technologies.

**Generating the dual expression vector**. The pFHL-plasmid for dual expression was constructed using four DNA fragments: (i) the Kanamycin resistance gene and ColE1 origen of replication from a C1 plasmid (Addgene plasmid #54842), followed by (ii) the dual promoter region from a pDuEx plasmid (pDress were the mTurquoise2, large spatial linker and P2A sequences were removed from the plasmid using NheI restriction sites[52]), (iii) the sequence coding for R-GECO1 including TorA-tag from pTorPE-R-GECO1 (a gift from Robert Campbell (Addgene plasmid #32465; http://n2t.net/addgene:32465; RRID:Addgene_32465), and (iv) the terminator region from a pDuEx plasmid. The four DNA fragments were generated by PCR amplification with Phusion High-Fidelity DNA Polymerase (Thermo Fisher Scientific) (primers 1–8). The DNA fragments were assembled using Gibson assembly[53]. *E. cloni* was transformed with the Gibson mix and correct construction was verified by digestion analysis and sequencing of the plasmid.

**Engineering of Tq-Ca-FLITS**. cpTq2 variants were constructed by PCR amplification from a tandem construct containing two mTurquoise2 proteins connected by a flexible GGSGG-linker (primers 9–26). mApple in R-GECO1 on pFHL-R-GECO1 was replaced with different cpTq2 variants by digestion of the vector and PCR fragments with SacI and MluI, followed by ligation. To create a library of

sensors with different linker lengths, a similar approach was taken, but now a library of cpTq2 fragments was generated by PCR using a mix of primers (primers 14–21 and 27). To create mutations at positions 146 and 150 of the FP (mTq2 numbering), again the same approach was taken, now using primers containing altered or degenerate codons (primers 28–32).

Mutations were also made at position 150 of regular mTurquoise2, by PCR amplification with primers containing a degenerated codon (primers 33–34). The PCR mix was DpnI digested and used for transformation.

**Plasmids for expression in mammalian cells**. The sensor sequence was amplified by PCR (primers 35–37) to generate cytoplasmic, nuclear, membrane, and Golgi-targeted versions of Tq-Ca-FLITS for transient transfection of mammalian cells. DNA fragments and vectors carrying the desired tag were digested with AgeI and BsrGI, followed by ligation. Of the following vectors the FP was exchanged for Tq-Ca-FLITS: 3xnls-mTurquoise2 (Addgene plasmid #98817) for a nuclear tag, pLck-mVenus-C1 (Addgene plasmid #84337) for a membrane tag, pmScarlet_Giantin_C1 (Addgene plasmid #85048) for a Golgi tag and mVenus-N1 (color variant of Addgene plasmid #54843) for an untagged version.

To generate stable cell lines, we used either the piggyBac transposase system or lentiviral transduction. To this end, Tq-Ca-FLITS including the 3xnls sequence was cloned into a PiggyBac vector containing a Puromycin resistance gene, by digestion with EcoRI/NotI and ligation, yielding pPB-3xnls-Tq-Ca-FLITS. The lentiviral plasmid for the doxycyclin-inducible expression of Tq-Ca-FLITS in organoids was adapted from the previously described pInducer20 × NLS-mKate2-P2A-HRAS[N17] [54]. The following replacements were made via In Fusion HD cloning kit (Clontech Laboratories): NLS-mKate2 reporter fluorophore was replaced by PCR encoding NLS-mMaroon (primers FW 5′-actagtccagACGCGtCcaccATGCcaaagaagaaacggaagg taggatcaatggtgagcaagggcgag-3′, RV 5′-ttgtacagctccGTTAACccattaagtttgtgccccagtttgc-3′); HRAS[N17] was replaced by PCR encoding Tq-Ca-FLITS (primers FW 5′-CCCT GGACCTGCTAGCatggtgagcaagggcgag-3′, RV 5′-gccctctagactcgagCTA CTTCGCTGTCATCATTTGGACAAAC-3′); ires-puro resistance cassette was replaced by ires-blast (primers FW 5′-taaggatccgcgggccGCATCGATGCCTAGTGC CATTTGTTCAGTG-3′, RV 5′-tctagagtcgcgggccgcCATGCATTTAGCCCTCCCACA C-3′). The resulting plasmid is indicated as pInducer-NLS-mMaroon-P2A-3xnls-Tq-Ca-FLITS.

We also constructed a plasmid for lentiviral transduction for normal expression of Tq-Ca-FLITS. Here In-Fusion Cloning was used to insert the coding sequences of H2B-mMaroon and 3xnls-Tq-Ca-FLITS connected with a P2A sequence in a lentiviral vector. The resulting plasmid is indicated as pLV-H2B-Maroon-P2A-3xnls-Tq-Ca-FLITS.

**Bacterial screening**. *E. cloni* bacteria were used for two screening methods.

(i) Bacterial test. Bacteria expressing a sensor variant with the TorA-tag were grown overnight (O/N) in Luria-Bertani medium (LB, 10 g/L Bacto Tryptone, 5 g/L Bacto Yeast extract, 10 g/L NaCl). The bacterial suspension was pipetted in triplicate in a CELLSTAR 96-wells plate with black walls (655090, Greiner-Bio). Intensity was recorded at room temperature (RT) using a FL600 microplate fluorescence reader controlled by KC4™ software (Bio-Tek) with 430/25, 485/20, or 555/25 nm excitation and 485/40, 530/25, or 620/40 nm emission for, respectively, CFP, GFP, or RFP, and averaging each well 10×. Intensity was again recorded after addition of 0.5 mM EDTA or MilliQ water (control). Intensities were divided by a well with 1 mg/ml Erythrosin B (EB), and a background (clear LB) was subtracted. Finally, wells were normalized to the first read of the same well to obtain the intensity fold-change $F_{max}/F_0$.

(ii) Periplasm test. Alternatively, the periplasmic shock fluid containing the sensor was isolated, using a cold osmotic shock protocol as described before[6]. Intensity of periplasmic fluid was measured before and after addition of 0.1 mM CaCl$_2$ or MilliQ water, using the microplate reader described above. Intensities were divided by a well with 1 mg/ml EB, background (clear buffer) was subtracted and wells were normalized to the first read of the same well to obtain the intensity fold-change $F_{max}/F_0$. Each periplasmic isolate was measured in duplicate.

**HeLa cell culture**. HeLa cells (CCL-2) acquired from the American Tissue Culture Collection were maintained in full medium, Dulbecco's modified Eagle medium + GlutaMAX (61965, Gibco) supplemented with 10% fetal bovine serum (FBS) (10270, Gibco), under 7% humidified CO$_2$ atmosphere at 37 °C. Cells were washed with Hank's buffered salt solution (HBSS) (14175, Gibco) and trypsinized (25300, Gibco) for passaging. No antibiotics were used unless otherwise stated.

HeLa cells were grown on round coverslips (Menzel, no. 1, 24 mm diameter, Thermo Fisher Scientific) in a six-well plate for imaging. Transfection mixture was prepared in Opti-MEM (31985047, Thermo Fisher Scientific) with 2.25 μg Polyethylenimine in water (PEI, pH 7.3, 23966, Polysciences) and 250 ng plasmid DNA, and incubated for 20 min before addition to the cells. The coverslips were 1 or 2 days post-transfection mounted in an AttoFluor cell chamber (A7816, Thermo Fisher Scientific) and microscopy medium (137 mM NaCl, 5.4 mM KCl, 1.8 mM CaCl$_2$, 0.8 mM MgSO$_4$, 20 mM D-Glucose, 20 mM HEPES pH 7.4) was added.

**Stable expression of Tq-Ca-FLITS in HeLa**. HeLa cells were transfected with pPB-3xnls-Tq-Ca-FLITS using PEI as transfection agent. One day post transfection, transfected cells were selected by addition of puromycin (1 μg/ml) to the medium for 24 h. The remaining cells were expanded for fluorescence-assisted cell sorting (FACS). Briefly, cells were washed, trypsinized, resuspended in full medium, and spun down at 1000 r.p.m. for 4 min. Cells were washed twice in HF (2% FBS in HBSS) and resuspended in an appropriate volume of HF. The cell suspension was filtered through a 70 μm filter. Cells were sorted into full medium supplemented with P/S (100 U/ml penicillin and 100 μg/ml streptomycin) and 25 mM HEPES (pH 7.4) on a BD FACSARIA3, with 407 nm excitation, and 502 nm long-pass and 510/50 nm band-pass emission filters. The sample chamber and collection devices were set at 4 °C for increased cell survival.

Single cells were gated based on forward and side scatter. Live-cell gating with 4′,6-diamidino-2-phenylindole was not possible due to spectral overlap with the fluorescence of Tq-Ca-FLITS. The positive gate for Tq-Ca-FLITS was determined based on untransfected wild-type HeLa cells. Cells were sorted into a high (34% of positive events) and low (66%) fluorescent pool. FACS data were analyzed with FlowJo. After sorting, cells were cultured with P/S for several weeks or until freezing down. Cells from the high pool were used for in situ calibration of Tq-Ca-FLITS.

**Lifetime imaging**. Fluorescence lifetime was recorded at RT with a 15–20 s interval, before and after addition of a mix of ionomycin (10 μg/ml, I-6800, LClaboratories) and calcium (5 mM). Two frequency domain FLIM microscopes were used.

(i) A home-build Zeiss setup controlled by Matlab 6.1 software, composed of an Axiovert 200 M inverted fluorescence microscope (Zeiss) with a II18MD modulated image intensifier (Lambert Instruments) coupled to a CoolSNAP HQ CCD camera (Roper Scientific) and two computer-controlled HF-frequency synthesizers (SML 01, Rohde & Schwartz), one driving the intensifier and the other driving a 440 nm modulated laser diode (PicoQuant, LDH-M-C-440) through a MDL-300 driver unit[55]. The excitation light is modulated at 75.1 MHz and reflected by a 455 nm dichroic mirror onto the sample. Emission is filtered with a 480/40 nm band-pass emission filter. A ×40 (Plan NeoFluar NA 1.3 oil) objective was used.

(ii) A Lambert Instruments FLIM Attachment (LIFA) setup composed of an Eclipse Ti microscope (Nikon) with a Lambert Instruments Multi-LED for excitation, a LI$^2$CAM camera and a LIFA signal generator (all Lambert Instruments) to synchronize the light source and the camera. For CFP excitation, a 446 nm light-emitting diode (LED) was used, combined with a 448/20 nm excitation filter, a 442 nm dichroic mirror, and a 482/25 nm band-pass filter. For RFP excitation, a 532 nm LED was used, combined with a 534/20 nm excitation filter, a 561 nm dichroic mirror and a 609/54 nm band-pass filter (all filters from Semrock). Alexa488 or EB was used as a reference to calibrate the instrumentation, with a known mono-exponential lifetime of 4.05[19,56,57] or 0.086 ns[58–60], respectively. Cells were imaged using a ×40 (Plan Apo, NA 0.95 air) or a ×60 (Plan Apo, NA 1.40 oil) objective.

Data from the Zeiss setup were analyzed as described before[61]. Data from the LIFA setup were converted to lifetime images by the LI-FLIM software (version 1.2.13).

Regions of interest were selected to extract the average lifetime. Only cells with appropriate average intensity were selected to avoid influence of background fluorescence (>200 for the Zeiss setup, >2000 for the LIFA setup). The in situ fold-change $F_{max}/F_0$ of Tq-Ca-FLITS was determined from the intensity data of a FLIM stack, corrected for background intensity.

**Ratiometric imaging**. Fluorescent ratio imaging was with two different microscopes at 37 °C. (i) An Eclipse Ti microscope (Nikon) equipped with an Intensilight C-HGFIE (Nikon) for excitation and an Orca-Flash4.0 camera (Hamamatsu). Cells were imaged using a ×40 (Plan Apo, NA 0.95 air) objective. For FRET imaging of YCaM3.60 we used a 448/20 nm excitation filter combined with a 442 nm dichroic mirror and a 482/25 nm emission filter for the donor, and a 514 nm dichroic mirror and a 542/27 nm emission filter for the acceptor. For MatryoshCaMP6s, we used a 448/20 nm excitation filter, a 488 nm dichroic mirror, and a 520/30 nm emission filter for imaging of the GFP, and a 448/20 nm excitation filter, a 561 nm dichroic mirror, and a 609/54 nm emission filter for LSSmOrange (all filters from Semrock).

(ii) An Axiovert 200 M inverted fluorescence microscope (ZEISS) equipped with an Intensilight C-HGFIE (Nikon) for excitation and a CoolSNAP HQ CCD camera (Roper Scientific). Cells were imaged using a ×40 (Plan Neofluar, NA 1.30 oil) objective. YCaM3.60 was excited with 420/30 nm. A 455 nm dichroic mirror was used. CFP fluorescence was collected at 470/30 nm and YFP fluorescence at 535/30 nm. MatryoshCaMP6s was excited with 440/30 nm followed by a 490 nm dichroic mirror. GFP fluorescence was collected at 525/40 nm and LSSmOrange fluorescence at 600/37 nm.

Background intensity was subtracted from the images and a ratio image was calculated using ImageJ (version 1.52k). The average intensity of each channel and the ratio was determined for individual cells.

**Protein isolation**. His-tagged Tq-Ca-FLITS, jGCaMP7c, RCaMP1h, and mTurquoise2 were isolated from bacterial culture essentially as described before, using Ni$^{2+}$ loaded His-Bind resin[62]. In the final step, the isolated protein was overnight dialyzed in 10 mM Tris-HCl pH 8.0. No further purification was performed.

**Quantum yield**. Purified Tq-Ca-FLITS and jGCaMP7c were diluted 10× in 10 mM Tris-HCl with 100 μM CaCl$_2$ or 5 mM EGTA, for the calcium-bound or -unbound state, respectively. The absorbance spectra were measured with a spectrophotometer (Libra S70, Biochrom) between 260 and 650 nm for CFP and between 260 and 700 nm for GFP (step size 1 nm, bandwidth 2 nm). Buffer without protein was used as reference. Each dilution was measured three times. Three dilutions were made using the initial dilution, with an absorbance at 440 nm ($A_{440}$) of $0.002 < A_{440} < 0.02$, each in triplicate. Emission and excitation spectra were recorded with a LS55 fluorimeter controlled by FL WinLab software (Perkin Elmer), with a step size of 0.5 nm and a scan speed of 200 nm/min and using buffer as reference. Emission was recorded at 450–650 nm (2.5 nm slit) with 440 nm excitation (4 nm slit). Excitation spectra were recorded at 250–490 or 250–530 nm (2.5 nm slit) measuring emission at 500 or 540 (4 nm slit), for CFP and GFP, respectively.

Absorbance spectra were corrected by subtraction of the offset of the spectrum between 631 and 650 nm. Emission spectra were corrected for spectral sensitivity of the detector. The spectral area ($I_{em}$) under corrected emission spectra was calculated by integration between 450 and 650 nm. The $A_{440}$ was plotted versus the "$I_{em}$" and the slope "$s$" was determined while forcing the regression line through the origin, $A_{440} = s \times I_{em}$. The QY was determined using mTurquoise2 as reference with a known QY of 0.93 (Eq. 1).

$$QY_s = QY_r \times \frac{s_s}{s_r} \qquad (1)$$

Subscripts "s" and "r" indicate the sample and the reference, respectively. The average emission and excitation spectra were calculated from the highest protein concentration.

**Extinction coefficient**. Purified Tq-Ca-FLITS was 4× diluted in calcium buffers containing 0 or 39 μM free calcium of the Calcium Calibration Buffer Kit #1 (C3008MP, Thermo Fisher Scientific). The absorbance spectra were measured before and >5 min after addition of 1 M NaOH, at 260–650 nm with 1 nm step size and 1 nm bandwidth. Corresponding buffer was used as reference. The concentration of unfolded protein was determined using the Beer–Lambert law and assuming an extinction coefficient ($\varepsilon$) at 462 nm of 46 mM$^{-1}$ cm$^{-1}$ for the free cyan chromophore[63]. Next, $\varepsilon$ was determined at 440 nm for the calcium-free and -bound states. The average absorbance spectra of three measurements were plotted.

**In vitro calibration**. Purified Tq-Ca-FLITS and jGCaMP7c were diluted 100× in calcium buffers ranging from 0 to 39 μM, using the Calcium Calibration Buffer Kit #1 according to manufacturers' instructions. Dilutions were made in triplicate. Fluorescence was measured at RT in a 96-well plate with black walls and flat glass bottom (89626, Ibidi) using a microplate fluorescence reader with settings as described under "bacterial screening." The fluorescence "$f$" was fit to the Hill equation to determine the $K_d$, using the Nonlinear Least Squares method of the R Stats Package (version 3.3.3) in R Studio (version 1.0.136) with default settings (Eq. 2).

$$f = f_{min} + \frac{(f_{max} - f_{min})}{\left(\frac{K_d}{L}\right)^n + 1} \qquad (2)$$

Subscripts "max" and "min" indicate the maximum and minimum fluorescence, "$K_d$" the microscopic dissociation constant, "$L$" the known free Ca$^{2+}$ concentration and "$n$" the Hill coefficient.

The lifetime of each well in the same 96-well plate was recorded at RT using the LIFA setup described earlier, using the ×40 (Plan Apo, NA 0.95 air) objective. Recorded sample stacks and a reference stack were converted into lifetime images by an ImageJ macro[52,64].

The average phase and modulation lifetime ($\tau_\varphi$ and $\tau_M$) of the full view were extracted. Both lifetimes were separately fitted with the Hill equation (Eq. 2), with "$\tau$" instead of "$f$," using the Nonlinear Least Squares method of the R Stats Package (version 3.3.3) in R Studio (version 1.0.136) with default settings. The Phase ($\Phi$) and modulation ($M$) were calculated from the recorded lifetimes and displayed in a polar plot as $G$ and $S$ coordinates (Eq. 3).

$$\Phi = atan(\omega\tau_\varphi) \text{ and } M = \sqrt{\frac{1}{1 + (\omega\tau_M)^2}} \qquad (3)$$

$$G = M\cos(\Phi) \text{ and } S = M\sin(\Phi)$$

The angular frequency of modulation, $2\pi f$, is given by "$\omega$." Measurements were projected on the straight line between the two extremes (min and max) and

converted to line fraction "$a$" (Eq. 4).

$$dG = G - G_{min} \text{ and } dS = S - S_{min}$$
$$a = \frac{dG \times dG_{max} + dS \times dS_{max}}{dG_{max}^2 + dS_{max}^2} \qquad (4)$$

The line fraction was corrected of for the intensity contribution of the two states to find the true fraction "$F$," with $F = 1$ representing all sensors in the calcium-bound state (Eq. 5). The intensity ratio ($R$) between states in vitro was determined to be $R = 3.51$ at pH 7.0, based on $f_{max}/f_{min}$ from the pH sensitivity experiments (see below).

$$F = \frac{a}{R \times (1 - a) + a} \qquad (5)$$

The fraction ($F$) was fitted with the Hill equation (Eq. 2) with "$F$" instead of "$f$" to find the in vitro $K_d$, using the Nonlinear Least Squares method of the R Stats Package (version 3.3.3) in R Studio (version 1.0.136), using the port algorithm.

**In situ calibration**. All steps were performed at 37 °C. Calcium buffers ranging from 0 to 39 μM (11 concentrations, in triplicate) were prepared using the Calcium Calibration Buffer Kit #1. HeLa cells stably expressing nuclear-targeted Tq-Ca-FLITS were grown on coverslips and mounted in a cell chamber as described above. Cells were washed twice with HBSS without calcium (14175, Gibco) and calcium buffer was added. Cells were incubated for 15–20 min with 4 μg/ml rotenone and 1.8 mM 2-deoxy-D-glucose. Lifetime stacks were recorded using the LIFA setup described under "lifetime imaging," while adding 10 μM of digitonin. A ×40 (Plan Apo, NA 0.95 air) objective was used and fluorescence lifetime was recorded every 20 s. Recorded data were converted into lifetime images by an ImageJ macro[52,64].

The average lifetimes of all pixels in a view with intensity >1000 were plotted over time (~30–50 cells per view). Equilibrium was reached (reaction to digitonin) after >6 min and the corresponding lifetimes were listed. Next, the same approach to determine the in situ $K_d$ was taken as for the in vitro data, using $R = 3.02$ as determined from $F_{max}/F_0$ from HeLa cells stimulated with ionomycin and calcium.

In addition, line fraction "$a$" was fitted with the Hill equation (Eq. 2), with "$a$" instead of "$f$", to determine parameters for conversion of experimental data to calcium concentrations, without the need to correct for intensity contribution of the calcium-free and -bound states.

The sample standard deviation of the fraction "$F$" was calculated for the highest and lowest calcium concentration, 0 and 39 μM. From this, we determined the 95% confidence interval (CI). To determine the lowest fraction we can reliably measure, we added the 95% CI to the mean of the 0 μM calcium measurements. To determine the highest measurable fraction, we did a subtraction.

**pH sensitivity**. A series of buffers ranging from pH 2.8 to 10.0 were prepared, using 50 mM citrate buffer (pH 2.8–5.8), MOPS buffer (pH 6.3–7.9), and glycine/NaOH buffer (pH 8.3–10.0). Buffers additionally contain 0.1 M KCl and 0.1 mM CaCl$_2$ or 5 mM EGTA. The pH of each buffer was determined including all components shortly before use. Purified Tq-Ca-FLITS, jGCaMP7c and RCaMP1h was diluted 100× in the prepared buffers.

Fluorescence was measured in triplicate in a 96-well plate with black walls and flat glass bottom (89626, Ibidi) using a FL600 microplate fluorescence reader controlled by KC4™ software (Bio-Tek) with settings as described under "bacterial screening." The fluorescence intensity ($f$) was corrected for background and fit to the Henderson–Hasselbalch equation (Eq. 6) using the Nonlinear Least Squares method of the R Stats Package (version 3.3.3) in R Studio (version 1.0.136), using the port algorithm restricted to $f_{min} \leq 0$. The calcium-bound state of Tq-Ca-FLITS did not fit to this model, therefore a model with two p$K_a$ values was applied (Eq. 7).

$$f = f_{min} + \frac{(f_{max} - f_{min})}{1 + 10^{n(pK_a - pH)}} \qquad (6)$$

$$f = f_{min} + \frac{(f_{med} - f_{min})}{1 + 10^{n_1(pK_{a,1} - pH)}} + \frac{(f_{max} - f_{med})}{1 + 10^{n_2(pK_{a,2} - pH)}} \qquad (7)$$

The "$n$" indicates the Hill coefficient, "p$K_a$" the apparent p$K_a$, and "min," "med," and "max" the minimum, a medium, and the maximum fluorescence. The model for the Ca$^{2+}$ saturated state was divided over the model of the Ca$^{2+}$-free state to gain the dynamic range.

The lifetime of each well in the same 96-well plate was recorded at RT similar to the in vitro calibration. Recorded data was converted into lifetime images by the LIFLIM software (version 1.2.13). The average phase and modulation lifetime ($\tau_\varphi$ and $\tau_M$) of the full view were extracted.

**Magnesium sensitivity**. A series of buffers was prepared at pH 7.1 and 20 °C, each designed to contain a specific concentration of free magnesium ions (0, 0.9, 1.8, 3.7, 9.3, and 18.7 mM) and either 0 or 1 mM free calcium, as calculated using the program "Ca-Mg-ATP-EGTA Calculator v1.0 using constants from NIST database #46 v8." The buffers including calcium contain each 1 mM CaCl$_2$, 20 mM HEPES pH 7.1 and 0, 0.9, 1.8, 3.7, 9.3, or 18.7 mM MgCl$_2$. The calcium-free buffers contain each 3 mM EGTA, 20 mM HEPES pH 7.1, and 0, 1, 2, 4, 10, or 20 mM MgCl$_2$. The

expected trace of calcium in the calcium-free buffers is 1 nM or less. The pH was measured after mixing of the buffers and checked again before use.

Purified Tq-Ca-FLITS and jGCaMP7c were 100× diluted in the buffers and the intensity was recorded similar as done for the pH measurements. The dynamic range was calculated as the average intensity of the calcium-bound state divided by the calcium-free state. The lifetime of Tq-Ca-FLITS was recorded and processed in the same manner as described under "pH sensitivity."

**EC culture and TEM**. Primary human umbilical vein endothelial cells (HUVECs) acquired from Lonza (P1052, Cat #C2519A) were maintained in culture flasks pre-coated with fibronectin (FN, 30 μg/mL, Sanquin) in EGM-2 medium supplemented with SingleQuots (CC-3162, Lonza) and P/S, under 5% humidified $CO_2$ atmosphere at 37 °C. HUVECs were passaged by washing twice with phosphate buffered saline (PBS) and trypsinization. Trypsin was inactivated with Trypsin Neutralization Solution (CC-5002, Lonza). HUVECs were transfected by microporation at passage #4 with 2 μg plasmid DNA containing the Lck version of Tq-Ca-FLITS. For microporation, the Neon Transfection System (MPK5000 Invitrogen) and corresponding Neon transfection kit were used according to the manufacturers' protocol. We used the R buffer from the kit, the 100 μL tips, and a 30 ms pulse of 1300 V. After microporation, HUVECs were directly seeded on FN-coated round coverslips for imaging, similar as described for HeLa cells. For TEM experiments, the buffer was removed by centrifugation ($200 \times g$, 3 min, RT) and HUVECs were seeded in a FN-coated μ-Slide VI 0.4 (80606, Ibidi).

Polymorphonuclear cells, consisting mainly of neutrophils, were isolated from whole blood from healthy donors (Sanquin), stored O/N at RT. Briefly, blood was diluted 1 : 1 in PBS with 5% (v/v) trisodium citrate and pipetted on top of 12.5 mL Percoll (1.047 g/mL) at RT. Cells were centrifuged at $800 \times g$ for 20 min (slow start, low brake) and all fractions, except neutrophils and erythrocytes, were removed. Erythrocytes were lysed twice in ice-cold isotonic lysis buffer (155 mM $NH_4Cl$, 10 mM $KHCO_3$, 0.1 mM EDTA). Remaining neutrophils were washed with PBS, followed by centrifugation at $450 \times g$ for 5 min. Neutrophils were resuspended in HEPES medium (20 mM HEPES, 132 mM NaCl, 6 mM KCl, 1 mM $MgSO_4$, 1.2 mM $K_2HPO_4$, 1 mM $CaCl_2$, 0.1% D-glucose, 0.5% Albuman (Sanquin Reagents)) and kept at RT for 4 h maximum. Neutrophils were prior to use labeled with 2 μM Calcein Red-Orange dye for 20 min at 37 °C. Dye was removed by centrifugation and the neutrophils were directly used. Neutrophils were isolated from four donors.

Two days post transfection, μ-Slides containing HUVECs were stimulated with 10 ng/ml tumor necrosis factor-α. The slides were connected after 4 h to a closed perfusion system to mimic the blood flow. Cells were exposed to flow rates of 0.8 dynes/cm$^2$ (HEPES buffer) and kept under 5% humidified $CO_2$ atmosphere at 37 °C. Neutrophils where injected in the closed system to allow TEM. Then, 100 μM histamine or a mix of 10 μg/ml ionomycin and 5 mM calcium was used as positive control for the sensor.

Fluorescence lifetime was measured using the LIFA setup described under "lifetime imaging," with a ×40 (Plan Apo, NA 0.95 air) objective. Lifetime stacks from the cyan channel were recorded every 13.5 s, alternated with an image in the red channel. For red excitation, a 532 nm LED was used, combined with a 534/20 nm excitation filter, a 575 nm dichroic mirror, and a 609/54 nm band-pass filter.

A background correction for removal of background lifetime was done on the lifetime stacks using a manually indicated background region. The background corrected data were used to calculate for each pixel the polar coordinates $M$ and $\Phi$ that represent the phase and modulation lifetimes[64]. The polar coordinates were corrected for daily variance compared to the in situ calibration. To this end, the position of the recorded high lifetime state of the positive controls was forced to the position of the high lifetime state of the in situ calibration. The corresponding correction factors (addition for "$\Phi$" and division for "$M$") were used to correct all recorded TEM data. Line fraction "$a$" was calculated and converted into the calcium concentration using the in situ calibration.

Neutrophils were manually tracked in ImageJ (version 1.52k) to collect their intensities. The calcium concentration in the underlying HUVEC was extracted, as well as the cyan intensity. The stage of migration was manually assigned (crawling, diapedesis or post-diapedesis). As diapedesis is a relatively rapid process, this yields a relatively low number of data points. We also measured the concentration before adhesion at the position where a neutrophil would adhere at a later timepoint and assigned the stage "before".

**Organoid imaging**. The study was approved by the UMC Utrecht (Utrecht, The Netherlands) ethical committee and was in accordance with the Declaration of Helsinki and according to Dutch law. This study is compliant with all relevant ethical regulations regarding research involving human participants.

Human intestinal organoids from the human ileum (N39; https://huborganoids.nl) stably expressing NLS-mMaroon and nuclear-targeted Tq-Ca-FLITS upon doxycycline induction were generated by lentiviral transduction and maintained as described elsewhere[65]. For differentiation towards EECs, organoids were treated with 1 μg/mL doxycycline for 48 h, five days after seeding. Four days later (day six of differentiation), organoids were analyzed. Then, 10 μg/ml GPBAR-A (4478, Bio-Techne, Abingdon, UK) was added to the organoids while recording

the fluorescence lifetime, using the LIFA setup described under "lifetime imaging," with a ×40 (Plan Apo, NA 0.95 air) objective. Lifetime stacks were recorded every 10 s. For positive control, we added 0.25% (w/v) Triton X-100 together with 12.5 mM $CaCl_2$.

Datasets were used to calculate for each pixel the polar coordinates $M$ and $\Phi$ that represent the phase and modulation lifetimes[64]. The polar coordinates were corrected for daily variance compared to the in situ calibration. To this end, the position of the recorded high lifetime state of the positive controls was forced to the position of the high lifetime state of the in situ calibration. The corresponding correction factors (addition for "$\Phi$" and division for "$M$") were used to correct the data. Line fraction "$a$" was calculated and converted into the calcium concentration using the in situ calibration.

**Reporting summary**. Further information on research design is available in the Nature Research Reporting Summary linked to this article.

## Data availability
The data produced in this study are available within the article and its Supplementary Information. All raw data will be available at Zenodo.org upon publication, doi: 10.5281/zenodo.5649554. Plasmids are deposited for distribution through Addgene (www.addgene.org). The plasmids and corresponding addgene numbers are: pFHL-Tq-Ca-FLITS: #129628, 3xnls-Tq-Ca-FLITS: #129626, Lck-Tq-Ca-FLITS: #129627, pPB-3xnls-Tq-Ca-FLITS: #145030, and pLV-H2B-Maroon-P2A-3xnls-Tq-Ca-FLITS: #145027. Source data are provided with this paper.

## Code availability
The most up-to-date custom code and scripts are available through GitHub: https://github.com/Franka-van-der-Linden/Quantitative-Calcium-Imaging. A snapshot of this code is included in the repository https://doi.org/10.5281/zenodo.5649554.

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

## Acknowledgements

F.H.L. was supported by a NWO Chemical Sciences ECHO grant (711.017.003). E.M. was supported by a NWO ALW-OPEN grant (ALWOP.306). M.P. was supported by a NWO-TTP grant (14691). J.v.B. was supported by a ZonMW NOW Vici grant (91819632). The funders had no role in study design, data collection and analysis, decision to publish, or preparation of the manuscript.

## Author contributions

F.H.L. and J.G. conceptualized the project, designed the experiments, interpreted the results, and wrote the manuscript. F.H.L., E.K.M., J.A., and J.v.B. participated in the experiments on endothelial cells and were involved in isolation of leukocytes. J.P., J.D.B., B.P., and H.C. transduced and prepared organoids and assisted with experiments on organoids that were carried out by F.H.L. A.O.C. generated stable HeLa cell lines. S.M.A.M. performed FACS. F.H.L., T.W.J.G., and M.P. were involved in FLIM experiments, performed data analysis, and assisted with interpretation of the data. All authors approved the final manuscript.

## Competing interests

H.C. is an inventor on patents related to organoid technology. His full disclosure is given at: https://www.uu.nl/staff/JCClevers/Additionalfunctions. The remaining authors declare no competing interests.
