## [Peer Review File · Nature Communications]

Reviewers' Comments:

Reviewer #2:

Remarks to the Author:

The authors have addressed my comments well. For example, the added comparisons to existing red indicators, discussions of pH sensitivity and apo state brightness as a important factors for quantitative measurements, measurements of Mg²⁺ sensitivity, and general improvements to readability have substantially improved the manuscript.

I support publication of the paper in its current form, but I have two minor points to add:

I did not understand the statement on line 105:

"For the FRET-sensors this is caused by a close to 100% efficiency in energy transfer in one of the states of the sensor"

Why would that preclude a lifetime change? Is this a typo? I imagine that a FRET sensor going from e.g. 0 to 100% FRET efficiency could exhibit a large lifetime change.

The rightmost part of figure 1G is unclear from the text. Is this a histogram of the images on the left? Is it necessary to show this? I feel a simple colorbar would make more sense here.

Reviewer #3:

Remarks to the Author:

FLIM detection provides robustness against many experimental fluctuations that alternative approaches suffer from, thus desirable in calcium imaging currently dominated by intensimetric and ratiometric detection. Towards that end, this paper offers a novel calcium sensor that enables FLIM detection. The turquoise sensor is also spectrally shifted from existing green and red calcium sensors, making room for spectral multiplexing. However, the applications shown does not demonstrate the advantage of the sensor as advertised yet. Also, the relatively slow speed basically exempts the sensor from brain or heart studies, some of the main applications for calcium sensors. It is highly desirable to demonstrate this sensor in a killer application that best show the advantage of FLIM imaging, ideally brain/heart calcium detection in a moving animal in vivo, or at least some ex vivo experiments in a beating cardiac organoid. I recommend publishing this exciting sensor on Nature Communications after addressing some issues as listed below.

Major point:

1)Fig2F-G, what exactly does the sensor offer over intensimetric sensors that are orders of magnitude better in terms of sensitivity in this static organoid application? To best show case the FLIM, the author should try to put in perspective in very complex in vivo environments in behaving animals.

2)While it is always good to see a sensor of a different emission band, the blue-shifted spectrum is undesirable for live cell/animal imaging due to the phototoxicity by low wavelength excitation. This would eventually limit the application of the sensor, and should be discussed in the manuscript. However, the issue can be turned around by doing two-photon FLIM, which is something the author may consider adding. Alternatively, one could show the advantage of the spectrally shifted emission by doing multicolor calcium imaging.

3)Fig2C-E, here the author was trying to demonstrate the sensor against an unsettled question, however, the lack of ground truth in this case leaves the results less convincingly.

4)Fig1G-H, it seems like the author was trying to demonstrate calcium detection by expressing the sensor in the nucleus, then manipulating cytoplasmic calcium by permeabilization. However, why not directly express the sensor in the cytoplasm?

5)Cultured cell are not really "in vivo". Page 15, line 342, "in vivo calibration", however, the

authors actually used a HeLa cell line instead.

6) There are some unsubstantiated statements. For example, Page 4, Line 102: "most current GECIs ... show hardly any or no QY...Exceptions are RCaMP1h and jRCaMP1b". However, in Table S2, clearly K-GECO1 and R-GECO1 show 3-4 fold change of QY, which is even larger than the sensor from this study.

Minor point:

1) Abbreviations such as QY should be explained in the first appearance, and used afterwards, however, this is not the case (Line 103, Line 263). Similarly, N, sd, CV in Table S1.

2) Error bars are missing in many places, e.g. Fig. 1F, FigS10.

3) Line 66, some these effects, some "of" these effects?

4) Line 68, "which is turn is", do you mean "which in turn is"?

Response to reviewers - manuscript NCOMMS-21-23268-T

We were very pleased with the generally positive comments of the reviewers and their support to publish this work in *Nature Communications*. To adhere to the *Nature Communications* style, we have restructured the manuscript with a clear distinction between Introduction, Results & Discussion. We have moved some key data and figures from supplemental to the main text (i.e. figure 1 and 4). Finally, the comments of the reviewers helped us to further strengthen the manuscript. Below we reply in a point-by-point manner to all the issues that were raised.

An article file with tracked changes is uploaded to the tracking system as 'Supplementary Info'.

Reviewer #1 was unable to review the revised version of your manuscript so we asked for another reviewer to comment on their points:

1. the first point is similar to point (2) made by reviewer #3 regarding short wavelengths in that they suggest that some discussion should be added regarding imaging in animals and that demonstrating 2P FLIM would solve this issue. While fluorescence imaging with blue is possible, calcium imaging as a physiological approach, which is almost always performed in live animals/tissues, is much more sensitive to phototoxicity.

We agree with the general note that shorter wavelength probes are not ideal with respect to phototoxicity, but it is hard to make a general statement as the extent of phototoxicity will depend on the tissue and the imaging method. We note that *in vivo* imaging with Turquoise based probes has been successfully done on an ordinary confocal microscope (doi: 10.1186/s12860-014-0044-2). Nowadays, imaging techniques that inflict less photodamage (especially lightsheet imaging) are available and therefore mTurquoise based probes should not be disregarded.

The suitability of mTurquoise2 as a probe is further supported by imaging of mTurquoise2 during the cell cycle, which is a process that is sensitive to phototoxicity (doi: 10.1016/j.celrep.2017.05.022). Finally, the option of 2 photon excitation is a good suggestion.

We have added a discussion on the suitability of mTurquoise-based probes and the potential of 2P imaging:

"Imaging with blue excitation light generally increases the risk of phototoxicity, especially a problem for in vivo research, but we note that mTurquoise2 has been successfully used for imaging of developing Nematostella vectensis embryos on an ordinary confocal microscope. In addition, mTurquoise2 was imaged to study the cell cycle, a process sensitive to photodamage. Together, these observations suggest that probes based on mTq2 can be successfully used in vivo. The use of light sheet imaging or multiphoton excitation could, in addition to improving the aforementioned optical sectioning, reduce the risk of phototoxicity. Furthermore, the blue-shifted color of Tq-Ca-FLITS could be beneficial for multicolor imaging. For example, we have successfully used it in combination with mClover and mTomato."

2. "a lower calcium sensitivity compared to other sensors", this is apparently wrong, as jRCaMP1a is actually more sensitive than GCaMP6s at low calcium concentration (Dana et al., eLife, 2016).

In this section, we discuss the properties of RCaMP1h and jRCaMP1b, which have a (demonstrated) lifetime contrast. We agree that jRCaMP1a has a higher affinity, but we do not take this probe into account as it only has a QY fold change of 1.68-1.86 (doi: 10.7554/eLife.12727 and 10.1016/j.bpj.2019.04.007) and it is unclear whether it shows a change in lifetime. For comparison, RCaMP1h, jRCaMP1b and Tq-Ca-FLITS show a QY fold change of 4.62, 3.6 and 3.0 respectively.

Reviewer #1 was unable to review this version. The comments from the replacement reviewer to be addressed are in the second paragraph of this letter (above).

We thank the replacement reviewer for the comments to which we reply above.

Reviewer #2 (Remarks to the Author):

The authors have addressed my comments well. For example, the added comparisons to existing red indicators, discussions of pH sensitivity and apo state brightness as a important factors for quantitative measurements, measurements of Mg²⁺ sensitivity, and general improvements to readability have substantially improved the manuscript.

I support publication of the paper in its current form, but I have two minor points to add:

We are very happy that the Reviewer supports publication.

I did not understand the statement on line 105:

"For the FRET-sensors this is caused by a close to 100% efficiency in energy transfer in one of the states of the sensor"

Why would that preclude a lifetime change? Is this a typo? I imagine that a FRET sensor going from e.g. 0 to 100% FRET efficiency could exhibit a large lifetime change.

We agree that this needs a better explanation. In a 100% FRET situation, the quenching of the donor is complete and no photons are emitted and consequently, no lifetime can be measured. This is less of a problem in ratio imaging where in the 100% FRET state all the energy is transferred to the acceptor, which emits a signal. (Note that in a biological system there is always a mixture of sensors in the FRET-state and in the non-FRET-state, with the concentration of calcium determining the balance between the two. When measuring a donor lifetime, this will be a mix of the non-FRET-state and the unmeasurable 100% FRET-state, so you will always measure the lifetime of the non-FRET-state. In case of FRET-ratio imaging, both donor and acceptor fluorescence are present in the mixture of the two FRET states, so a FRET change is clearly observed.)

An explanation is added to the text: *"For the FRET-sensors this is caused by a close to 100% efficiency in energy transfer in one of the states of the sensor. In this state, the donor fluorescence is completely quenched, hence no photons are emitted, and no donor lifetime*

can be measured. However, FRET can still be observed by ratio imaging as the energy is fully transferred to the acceptor, increasing its fluorescence."

The rightmost part of figure 1G is unclear from the text. Is this a histogram of the images on the left? Is it necessary to show this? I feel a simple colorbar would make more sense here.

We agree with the suggestion and changed the figure accordingly.
(Note that this is figure 3C in the revised manuscript).

Reviewer #3 (Remarks to the Author):

FLIM detection provides robustness against many experimental fluctuations that alternative approaches suffer from, thus desirable in calcium imaging currently dominated by intensimetric and ratiometric detection. Towards that end, this paper offers a novel calcium sensor that enables FLIM detection. The turquoise sensor is also spectrally shifted from existing green and red calcium sensors, making room for spectral multiplexing. However, the applications shown does not demonstrate the advantage of the sensor as advertised yet. Also, the relatively slow speed basically exempts the sensor from brain or heart studies, some of the main applications for calcium sensors. It is highly desirable to demonstrate this sensor in a killer application that best show the advantage of FLIM imaging, ideally brain/heart calcium detection in a moving animal in vivo, or at least some ex vivo experiments in a beating cardiac organoid. I recommend publishing this exciting sensor on Nature Communications after addressing some issues as listed below.

We are glad that the reviewer sees the advantage of measuring calcium with lifetime contrast. Two points are raised:

1) The applications do not demonstrate the advantage of the sensor.

We respectfully disagree. FLIM is robust against changes in intensity fluctuations (as the reviewer mentions) enabling the direct translation of lifetimes into absolute calcium concentrations. This is demonstrated in figure 5 (previously figure 2), where we show that (i) intensity fluctuations do not affect the read-out of the probe (panel C,D,E) and that (ii) we can quantify calcium concentrations in complex tissue (panel F,G) by directly translating the lifetime information into an absolute calcium concentration. This would be much more complicated when a FRET probe would have been used. See for instance Ponsioen at al (<https://www.nature.com/articles/s41556-021-00654-5>) where calibration of each measurement was required and still this does not yield absolute quantification, but relative changes.

2) The relatively slow speed exempts the sensor from brain or heart studies.

We agree that a current limitation of our study is the relatively low temporal resolution. We want to stress that this is not due to the design of the probe, which is similar to GCaMP type probes. Instead, it is due to the lifetime imaging equipment, which requires more photons and hence longer acquisition times for accurate measurements. Also, we decided to measure a larger field of view (360x270 μm) to capture more TEM events or a complete

organoid. So, we sacrifice speed for accuracy and field of view. With the new probe, there is now a choice to do slower, absolute measurements or fast and qualitative measurements.

To demonstrate that we can go faster we pushed our system to the max speed, enabling calcium imaging at 1.6 seconds/frame. These results are added in figure S12. As discussed in the paper, there are microscopes available that enable sub-second lifetime measurements: *“Our setup is not suitable for fast switching of the filters required for alternating imaging of cyan lifetime and red fluorescence, and it suffers from substantial response and dead times. In the choice of setup for our experiments we sacrificed speed for accurate imaging of several ECs in one field of view. When imaging only the lifetime in the CFP channel we reached a temporal resolution with Tq-Ca-FLITS of 1.6 seconds. Lifetime imaging can be further pushed to sub-second resolution by choice of improved and faster lifetime microscopy techniques such as a FALCON or Stellaris systems or siFLIM.”*

We note that a hybrid method with intermittent calibration proposed by Dedecker and colleagues (<https://doi.org/10.1101/2020.10.29.360214>) is a very promising method to further speed up the measurements (and reduce photobleaching). A similar strategy could be used to speed up the calcium measurements with FLIM. We added text to discuss this: *“Another promising method for speeding up lifetime measurements is by using a hybrid method with intermittent calibration proposed by Dedecker and colleagues.”*

Major point:

1) Fig2F-G, what exactly does the sensor offer over intensimetric sensors that are orders of magnitude better in terms of sensitivity in this static organoid application? To best show case the FLIM, the author should try to put in perspective in very complex in vivo environments in behaving animals.

We do not agree that intensimetric sensors are orders of magnitude better for a number of reasons:

- 1) The sensitivity of the sensor described in our manuscript is very well comparable to other genetically encoded sensors in terms of calcium sensitivity. We measured a K_d of 265 nM for Tq-Ca-FLITS. For the GCaMP6 series a K_d of 144-375 nM is reported ([10.1038/nature12354](https://doi.org/10.1038/nature12354)) and for the GCaMP7 series 68-298 nM ([10.1038/s41592-019-0435-6](https://doi.org/10.1038/s41592-019-0435-6)).
- 2) The dynamic range observed in vitro for GCaMP-type sensors is usually not reproduced in organoids. The GCaMP6 and 7 series are reported to have an in vitro fold change of 22-300 ([DOI 10.1038/s41592-019-0435-6](https://doi.org/10.1038/s41592-019-0435-6)). The maximum $\Delta F/F_0$ change reported is ~ 6 for GCaMP6s, but it is typical to see lower values ([DOI: 10.1016/j.cell.2021.04.029](https://doi.org/10.1016/j.cell.2021.04.029), [DOI: 10.1021/acsptsci.9b00090](https://doi.org/10.1021/acsptsci.9b00090), [DOI: 10.1038/s41551-020-0539-4](https://doi.org/10.1038/s41551-020-0539-4)). For comparison, we have previously reported a $\Delta F/F_0 \sim 1$ for the turquoise probe in organoids ([10.1016/j.cell.2020.04.036](https://doi.org/10.1016/j.cell.2020.04.036)), which is lower, but not ‘orders of magnitude’.
- 3) The intensimetric sensors are pH sensitive, which complicates quantitative measurements in gut organoids, where intracellular pH can vary (<https://doi.org/10.3389/fcell.2021.618135>). Our sensor shows no sensitivity to pH in the physiological range.

4) Intensiometric probes do not permit quantitative measurements of absolute concentrations as we demonstrate for cells and organoids in the current manuscript.

The advantages of the new probe are that:

- it enables quantitative imaging of absolute calcium concentrations
- it is not affected by pH in the physiological range.

We also added text to highlight the advantages in the discussion:

“To conclude, Tq-Ca-FLITS is the first GECI that incorporates a novel sensing mechanism based on a conformational change that directly modifies only the fluorescence quantum yield and fluorescence lifetime of a fluorescent protein independent of FRET (and without the need for a second fluorescent protein), providing contrast independent of sensor concentration. Tq-Ca-FLITS has a calcium sensitivity comparable to other existing GECIs and shows no sensitivity to pH in the biological range. Together, these properties make the sensor ideally suited for quantitative determination of calcium concentrations.”

2) While it is always good to see a sensor of a different emission band, the blue-shifted spectrum is undesirable for live cell/animal imaging due to the phototoxicity by low wavelength excitation. This would eventually limit the application of the sensor, and should be discussed in the manuscript. However, the issue can be turned around by doing two-photon FLIM, which is something the author may consider adding. Alternatively, one could show the advantage of the spectrally shifted emission by doing multicolor calcium imaging.

The reason to generate a sensor based on mTurquoise was that it had the potential to generate lifetime contrast. We agree that this may not be ideal for in vivo imaging, but we note that in vivo imaging with Turquoise based probes has been successfully done (doi: 10.1186/s12860-014-0044-2). We added a discussion on the suitability of mTurquoise-based probes and the potential of 2P imaging: *“Imaging with blue excitation light generally increases the risk of phototoxicity, especially a problem for in vivo research, but we note that mTurquoise2 has been successfully used for imaging of developing Nematostella vectensis embryos on an ordinary confocal microscope. In addition, mTurquoise2 was imaged to study the cell cycle, a process sensitive to photodamage. Together, these observations suggests that probes based on mTq2 can be successfully used in vivo. The use of light sheet imaging or multiphoton excitation could, in addition to improving the aforementioned optical sectioning, reduce the risk of phototoxicity.”*

We like the suggestion of using the Turquoise probe in a multicolor experiment and this is exactly what we have previously used the Turquoise calcium probe for (doi: 10.1016/j.cell.2020.04.036). We added text to highlight that this probe is suitable for multicolor imaging with mClover and dTomato: *“Furthermore, the blue-shifted color of Tq-Ca-FLITS could be beneficial for multicolor imaging. For example, we have used it successfully in combination with mClover and mTomato.”*

3) Fig2C-E, here the author was trying to demonstrate the sensor against an unsettled question, however, the lack of ground truth in this case leaves the results less convincingly.

We agree that this is an unsettled question, and this is discussed in supplementary note S4, with ample references to the existing literature. Some papers report on calcium changes, others do not. We demonstrate that although intensity changes are observed, we see no calcium changes and with our method we can actually give an upper limit for the calcium concentration during the process. The majority of cells do not show calcium concentrations above 40 nM during transmigration. Hence, we can conclude that in our (in vitro) model system, increases of calcium concentrations are not required for transendothelial migration.

4) Fig 1G-H, it seems like the author was trying to demonstrate calcium detection by expressing the sensor in the nucleus, then manipulating cytoplasmic calcium by permeabilization. However, why not directly express the sensor in the cytoplasm?

We have generated nuclear tagged stable cell lines, since it simplifies segmentation. This is not problematic for the calibration since the permeabilization generates a homogeneous calcium concentration throughout the cells.

5) Cultured cells are not really "in vivo". Page 15, line 342, "in vivo calibration", however, the authors actually used a HeLa cell line instead.

This is a good point. We wanted to differentiate it from the analysis on purified protein (in vitro) and we will change the cellular data to in situ.

6) There are some unsubstantiated statements. For example, Page 4, Line 102: "most current GECIs ... show hardly any or no QY...Exceptions are RCaMP1h and jRCaMP1b". However, in Table S2, clearly K-GECO1 and R-GECO1 show 3-4 fold change of QY, which is even larger than the sensor from this study.

We thank the referee for pointing out this omission, we will change the text to: "*Exceptions are R-GECO1, K-GECO1, RCaMP1h and jRCaMP1b...*"

Minor point:

1) Abbreviations such as QY should be explained in the first appearance, and used afterwards, however, this is not the case (Line 103, Line 263). Similarly, N, sd, CV in Table S1. The explanation of QY was moved to the first appearance. The explanation of N, sd and CV in table S1 are explained in the caption of the table, at their first appearance.

2) Error bars are missing in many places, e.g. Fig. 1F, FigS10. In these figures (Fig. 1F is currently 3B) mentioned all individual data points are plotted. In our view adding error bars here will only clutter the figures and add no additional information or change the conclusions.

3) Line 66, some these effects, some "of" these effects? This was corrected, thank you for pointing it out.

4) Line 68, "which is turn is", do you mean "which in turn is"? Corrected.

Reviewers' Comments:

Reviewer #2:

Remarks to the Author:

Having read the authors' Response and the updated manuscript, I continue to support its publication in its current form.

Reviewer #3:

Remarks to the Author:

I recommend publishing this manuscript on Nature Communications without further revision. In addition, I have a few comments for the authors.

First, I understand the difficulty of doing in vivo studies, but I highly encourage the authors to carry out in vivo experiments in follow up studies. This is really about introducing lifetime based calcium sensors to the wider community, which would be good to both the community the authors as well.

Second, concerning the sensitivity (dynamic range), as the authors have noticed, there seems to be substantial performance decrease in the well characterized GCaMP sensors going from in vitro to in vivo. The point I was making is that in vivo studies with side by side comparison with GCaMP would be very helpful for people in the field to select the best sensor according to their applications, related to the first point.

Third, the authors mentioned in the response: "This is not problematic for the calibration since the permeabilization generates a homogeneous calcium concentration throughout the cells. ". This is not true, a lot of studies have shown calcium gradients in different compartments within the cell, including between the nucleus and the cytoplasm.

Response to reviewers - manuscript NCOMMS-21-23268A

We were very pleased with the support to publish our work in *Nature Communications*. Below we reply in a point-by-point manner to all the issues that were raised. An article file with tracked changes is uploaded to the tracking system as 'Supplementary Info'.

Reviewer #2 (Remarks to the Author):

Having read the authors' Response and the updated manuscript, I continue to support its publication in its current form.

Fantastic, thanks!

Reviewer #3 (Remarks to the Author):

I recommend publishing this manuscript on Nature Communications without further revision.

Fantastic, thanks!

In addition, I have a few comments for the authors.

First, I understand the difficulty of doing in vivo studies, but I highly encourage the authors to carry out in vivo experiments in follow up studies. This is really about introducing lifetime based calcium sensors to the wider community, which would be good to both the community the authors as well.

We are as enthusiastic as the reviewer about the proposed studies and we are seeking funding and collaborations to perform these experiments.

Second, concerning the sensitivity (dynamic range), as the authors have noticed, there seems to be substantial performance decrease in the well characterized GCaMP sensors going from in vitro to in vivo. The point I was making is that in vivo studies with side by side comparison with GCaMP would be very helpful for people in the field to select the best sensor according to their applications, related to the first point.

We fully agree. Side-by-side comparisons, as we have previously done for fluorescent proteins, are very valuable.

Third, the authors mentioned in the response: "This is not problematic for the calibration since the permeabilization generates a homogeneous calcium concentration throughout the cells. ". This is not true, a lot of studies have shown calcium gradients in different compartments within the cell, including between the nucleus and the cytoplasm.

We agree that gradients do exist in living cells. During the calibration procedure, we abolish cellular activity (and hence gradients) by inactivating metabolism with the inhibitors rotenone and 2-deoxy-D-glucose.